# Easily Implementable Dietary Improvement Using Recipes: Analytical Method Applied to a Specific Region in Japan

**DOI:** 10.3390/nu17101614

**Published:** 2025-05-08

**Authors:** Makoto Hazama, Akiko Oda, Tamae Shimawaki, Naohito Ito, Mari Maeda-Yamamoto, Jun Nishihira

**Affiliations:** 1Department of Medical Management and Informatics, Hokkaido Information University, Ebetsu 069-8585, Japan; m-hazama@do-johodai.ac.jp (M.H.);; 2Institute of Food Research, National Agriculture and Food Research Organization, Tsukuba 305-8642, Japan; marimy@affrc.go.jp

**Keywords:** easily implementable dietary improvement, nutrient complementarity, complementarity in habits, complementarity in recipes

## Abstract

**Background/Objectives:** Improving one’s diet for the purpose of promoting health is constrained by people’s ingrained eating habits, as the eating patterns that align with their established habits do not necessarily correspond to a healthy way of eating. In addressing the issue of improving diet while taking both health and habits into consideration, this paper focuses on three concepts of complementarity related to food combinations and presents an approach using Japanese data. **Methods**: This paper first organizes three concepts of complementarity related to food combinations. The three concepts of complementarity are formulated based on (i) nutrients, (ii) habits, and (iii) recipes. The definitions of measurement scales corresponding to these concepts are also discussed. **Results**: Using data from a specific region in Japan, we analyzed three types of complementarities related to food combinations across different gender and age groups. This approach enabled us to visualize the potential for dietary improvements and identify effective strategies tailored to each group. For instance, among females aged 50 and above in this region, it was found that recipes incorporating combinations such as (α) milk and pasta, (β) salmon and pasta, (γ) horse mackerel or sardines with carrots, and (δ) onions with taro can efficiently support dietary improvement for this demographic, building upon their established dietary habits. The differences in recommended recipes for each group are due to variations in nutrients that tend to be insufficient and differences in established eating habits. **Conclusions**: A combination of foods with high (i) complementarity of nutrients constitutes a healthy diet, whereas a combination of foods with low (ii) complementarity in habits reflects dietary behaviors that are not sufficiently practiced within the relevant sub-population at present. Meanwhile, combinations of foods with high (iii) complementarity in recipes can serve as tools to bridge the gap between healthy eating patterns and established dietary habits.

## 1. Introduction

The challenges people face when trying to improve their diet for the purpose of maintaining and enhancing health range from physiological and psychological to cultural and social factors [1,2]. Among these, individual self-control in practicing dietary improvement holds a significant position [3]. Easily implementable dietary improvements that consider both health and personal established habits are an important research topic. The purpose of this paper, in short, is to present a method for finding highly practical food improvement measures.

When it comes to managing dietary habits, there is a wide variety of approaches. Naturally, dietary habits are not determined by a single, one-time decision made by an individual. In the routines of daily life, food-related choices are often made—sometimes consciously, sometimes unconsciously, sometimes with satisfaction, and at other times, reluctantly. What governs the management of dietary habits is not self-evident. Generally speaking, dietary patterns are nothing but combinations of foods. However, the number of possible food combinations is enormous. With *n* food items, there are $2^{n}-1$ potential combinations. Therefore, rather than analyzing meal patterns through comprehensive combinations, this paper will focus solely on the perspective of pairwise food combinations to explore potential dietary improvements. Specifically, this paper proposes the following three complementarity concepts related to combinations of two foods when considering easily implementable and health-promoting dietary improvements: (i) complementarity of nutrients, (ii) complementarity in habits, and (iii) complementarity in recipes. In short, the complementarity of nutrients between two foods is high when they contain a large amount of different nutrients from each other. The complementarity of two foods in habits is higher if the combination is prevalent in one’s dietary patterns. The complementarity of two foods in recipes is higher if the combination appears more frequently as ingredients in the same recipe.

Using a framework with these three complementarity concepts, we can formalize the potential for dietary improvements. First, the set of food combinations with (i) high complementarity of nutrients corresponds to a healthy eating pattern. Within this set, the subset of combinations with (ii) high complementarity in habits represents the eating patterns already well-established by the individual. Therefore, the set of food combinations with (i) high complementarity of nutrients, but (ii) low complementarity in habits should be targeted for dietary improvements for this person. The issue is that these combinations, while healthy, do not match the individual’s established habits. An important point here is that an individual’s habits are malleable. Dietary habits are not fixed and can be influenced by factors such as culinary convenience, appeal to satiety, education, advertising, and social relationships [4,5,6,7]. In the set of food combinations behind the targeted dietary improvements, there is a gap between (i) high complementarity of nutrients and (ii) low habits complementarity. In such cases, recipes can be used as tools to bridge this gap.

A recipe is a technology for combining multiple food ingredients. Recipes known as traditional dishes have been refined over long periods and can be considered methods of combination that have been widely accepted by people. Even newly proposed recipes must appeal to people’s preferences. So-called “bizarre foods” are excluded from recipes. This paper assumes that consideration for people’s preferences is sufficiently reflected not only in existing recipes, but also in newly proposed ones. Therefore, if there are food combinations in the set with (i) high complementarity of nutrients and (ii) low complementarity in habits that show (iii) high complementarity in recipes, then recipes using these food combinations can facilitate easier dietary improvements.

Regarding the method for measuring the scales of these three concepts of complementarity based on data, one (iii) of the three can be statistically straightforwardly defined, another (ii) is indicated in existing research, and the remaining one (i) is discussed in detail in this paper. Data from both genders in their 20s to 70s living in a specific region of Hokkaido, Japan, will be used for the measurement of complementarities. Analyses will be conducted by gender and age group to identify the potential for dietary improvements through recipes within each group.

## 2. Materials and Methods

### 2.1. Key Concepts: Three Types of Complementarity Related to Food Combination

This paper demonstrates a method for exploring the potential for dietary improvements that impose minimal burdens on individuals, based on the relationships among three types of complementarities between combinations of foods: (i) nutrients, (ii) habits, and (iii) recipes. This subsection sequentially explains the formulation of these three complementarity concepts.

#### 2.1.1. Nutrient Complementarity

The nutrient complementarity between two foods *i* and *j* is defined based on the nutrient content vectors per fixed quantity of foods *i* and *j*, respectively $\left(x_{i},\;x_{j}\right)$. While the data used will be mentioned later, the measurement units for the elements of the nutrient content vectors differ (for example, calcium is measured in mg, and vitamin D in μg). Therefore, the normalization transformation is performed using the inverse matrix of the sample standard deviation matrix $S\equiv {\hat{\mathrm{Var}\left(x\right)}}^{1/2}$ of the nutrient content vectors $z_{i}\equiv S^{-1}x_{i}$.

This paper distinguishes between broad complementarity and narrow complementarity in the formulation of the concept of nutrient complementarity of $\left(z_{i},\;z_{j}\right)$. Broad complementarity of nutrients consists of the following three elements (an illustration to aid in the schematic understanding of the three components of nutrient complementarity, θ, η, and φ, is provided in Appendix A).

Narrow complementarity θ;Quantitative effect η;Symmetry concerning quantity φ.

Firstly, narrow nutrient complementarity, as a concept, refers to the degree to which the directions of two normalized nutrient content vectors $\left(z_{i},\;z_{j}\right)$ differ. For example, to simplify, consider only calcium and iron as nutrients. If food *i* contains a large amount of calcium, but only a small amount of iron, and food *j* contains almost no calcium, but a large amount of iron, then the narrow nutrient complementarity between food *i* and food *j* is high. Straightforwardly, as a measurement, narrow nutrient complementarity can be defined as the angle θ between the normalized nutrient content vectors of the two foods $\left(z_{i},\;z_{j}\right)$:(1)$$\theta \left(z_{i},\;z_{j}\right)\equiv \cos^{-1}\left(\frac{z_{i}^{\prime }z_{j}}{\left|z_{i}\right|\left|z_{j}\right|}\right).$$

Note that the upper limit of the range of the angle $\theta \left(z_{i},\;z_{j}\right)$ between the normalized nutrient content vectors is not necessarily orthogonal (π/2), while the range of the angle $\theta \left(x_{i},\;x_{j}\right)$ between the two original nutrient content vectors is from 0 to π/2. This is because the normalization $z_{i}=S^{-1}x_{i}$ occasionally transforms the orthogonal coordinates into oblique coordinates. The transformation using the inverse matrix of the sample standard deviation matrix $S^{-1}$ not only normalizes the differences in units of nutrients, but also adjusts for the correlations in nutrient content among nutrients. For example, in the set of foods included in the data for the analysis mentioned later, the correlation between the contents of retinol and folic acid is significantly positive, while the correlation between the contents of vitamin C and phosphorus is negative. In this case, normalization by $S^{-1}$ transforms the coordinates so that the angle between the retinol and folic acid axes becomes greater than a right angle, and the angle between the vitamin C and phosphorus axes becomes less than a right angle.

Secondly, quantitative effect represents the quantitative magnitude of nutrient contents in combined foods. In the aforementioned narrow nutrient complementarity, the difference in the coverage of nutrients between two foods is captured by the angle between the two normalized nutrient content vectors. However, the two vectors near the origin and the other two vectors far from the origin are not distinguished if their angle magnitudes are the same. The quantitative effect is naturally defined by the following formula as a measure to supplement this point:(2)$$\eta \left(z_{i},\;z_{j}\right)\equiv \sqrt{{\left|z_{i}\right|}^{2}+{\left|z_{j}\right|}^{2}}.$$

Note that the quantitative effect is greater when combining foods with high nutrient content, smaller when combining foods with low nutrient content, and intermediate when combining foods with high and low nutrient content.

Thirdly, symmetry concerning quantity represents the symmetry in nutrient contents between two combined foods. When foods with similar amounts of nutrient content are combined and when foods with asymmetric amounts of nutrient content are combined, the respective contributions of the two foods to the total nutrient content of the combination differ. In the latter combination, it is as if the food with higher nutrient content is being “free ridden” by the food with lower nutrient content. Symmetry concerning quantity, defined as follows, is a measure to supplement this point:(3)$$\varphi \left(z_{i},\;z_{j}\right)\equiv 1-\left|\frac{4}{\pi }\cos^{-1}\left(\frac{\left|z_{i}\right|}{\eta \left(z_{i},\;z_{j}\right)}\right)-1\right|.$$

Note that the symmetry concerning quantity $\varphi \left(z_{i},\;z_{j}\right)$ is designed such that it takes the maximum value of 1 when the angle of the point $\left(\left|z_{i}\right|,\;\left|z_{j}\right|\right)$ is π/4, and approaches the lower bound of 0 as the deviation from π/4 increases, that is, when it approaches either 0 or π/2.

This paper defines nutrient complementarity (the broad concept of nutrient complementarity) between a generic combination of two foods $\left(i,\;j\right)$ as a vector consisting of three elements: narrow complementarity $\theta =\theta \left(z_{i},\;z_{j}\right)$, quantitative effect $\eta =\eta \left(z_{i},\;z_{j}\right)$, and symmetry concerning quantity $\varphi =\varphi \left(z_{i},\;z_{j}\right)$. The two measures, η (Equation (2)) and φ (Equation (3)), preserve sufficient information to reconstruct the unordered set of the absolute values of the two normalized nutrient content vectors of the two foods $\left\{\left|z_{i}\right|,\;\left|z_{j}\right|\right\}$. Therefore, the information on $\left(\theta ,\;\eta ,\;\varphi \right)$, i.e., the nutrition complementarity of a combination of two foods, is equivalent to the information on $\left(\theta ,\;\left\{\left|z_{i}\right|,\;\left|z_{j}\right|\right\}\right)$.

While the Mahalanobis distance $d\left(x_{i},\;x_{j}\right)\equiv {\left(x_{i}-x_{j}\right)}^{\prime }{\hat{\mathrm{Var}\left(x\right)}}^{-1}\left(x_{i}-x_{j}\right)$ is generally a strong candidate for measuring the nutrient complementarity between two food combinations, measuring with three scales $\left(\theta ,\;\eta ,\;\varphi \right)$ instead of a single scale $d\left(x_{i},\;x_{j}\right)$ allows for more detailed supplementation. In fact, the Mahalanobis distance is represented as a function of the three scales used in this paper. The transformation using the inverse of the standard deviation matrix, $S^{-1}$, is adopted for reasons similar to those of the Mahalanobis distance, namely differences in units and consideration of the correlation structure. The reason this paper measures nutrient complementarity using three scales instead of relying solely on the Mahalanobis distance lies in the weaknesses of the Mahalanobis distance. For more details, refer to Appendix A. Furthermore, for the convenience of readers, the calculation procedure for nutrient complementarity using a numerical example is provided in Appendix A.

Although broadly defined complementarity of nutrients has three dimensions, for the sake of convenience in the subsequent analysis, we define a composite measure that reduces them to a single dimension as follows:(4)$$I\left(z_{i},\;z_{j}\right)\equiv \min\left(F_{\theta }\left(\theta \left(z_{i},\;z_{j}\right)\right),\;F_{\eta }\left(\eta \left(z_{i},\;z_{j}\right)\right),\;F_{\varphi }\left(\varphi \left(z_{i},\;z_{j}\right)\right)\right),$$
where $F_{\theta }$, $F_{\eta }$, and $F_{\varphi }$ are the cumulative distribution functions of each variable. This definition indicates that the condition $I\left(z_{i},\;z_{j}\right)\geq 1-\tau$ for some $\tau \in \left[0,\;1\right]$ means that all three measures, θ, η, and φ, are above the $\left(1-\tau \right)\times 100$th percentile in their respective distributions. Using the composite measure $I\left(z_{i},\;z_{j}\right)$, we can rank all food combinations in terms of the degree of complementarity of nutrients.

#### 2.1.2. Complementarity in Habits

The complementarity in habits between two foods *i* and *j* is defined as the relative frequency of simultaneous intake of these two foods in an individual’s dietary habits. Our previous study [8] described the joint distribution of Food Frequency Questionnaire (FFQ) data for two foods as a two-dimensional multinomial distribution by modeling the probability of the two foods being consumed together as a Bernoulli trial for each eating occasion. This paper also adopts the same framework of [8], defining the probability of two foods *i* and *j* being consumed simultaneously in one eating occasion $p^{habit}\left(i,\;j\right)$ as the complementarity in habits for those food combinations:(5)$$p^{habit}\left(i,\;j\right)\equiv \mathrm{Pr}\left(\left\{i,\;j\right\}\in Y|X\right),$$
where Y is the set of food items consumed by an individual in an eating occasion, and X represents covariates of the individual’s attributes and/or condition.

Here, it is beneficial to emphasize the distinction between habits and preferences. Even within the analytical framework of this paper, habits and preferences regarding combinations of two types of food can be explicitly distinguished. Our previous study [8] defined the preference for combinations as “excess combination propensity”, defined as $\delta \equiv \mathrm{Pr}\left(\left\{i,\;j\right\}\in Y|X\right)-\mathrm{Pr}\left(\left\{i\right\}\in Y|X\right)\times \mathrm{Pr}\left(\left\{j\right\}\in Y|X\right)$, adjusting for the frequency of individual food consumptions. However, in this paper, to focus on the combined intake of foods as a means of efficient nutrient intake, we consider the gross combination propensity, i.e., the relative frequency of combinations, as the habits for combinations. Dietary habits, which are one of the three dimensions analyzed in this paper, comprehensively encompass various underlying factors such as preferences, monetary costs, cultural elements, culinary convenience, and appeal to satiety.

#### 2.1.3. Complementarity in Recipes

The complementarity in recipes between two foods *i* and *j* is simply defined as the relative frequency of these two ingredients being included in the same recipe. That is,(6)$$p^{recipe}\left(i,\;j\right)\equiv \mathrm{Pr}\left(\left\{i,\;j\right\}\in W\right),$$
where W represents the set of ingredients in a randomly selected recipe.

According to Equation (6), the probability that two foods are used as ingredients in a recipe randomly selected from a set of recipes represents the complementarity of those two foods within the recipe. Therefore, complementarity in recipes is defined for each set of recipes. The set of recipes reflects the environment surrounding the overall diet and generally varies depending on factors such as regional food culture, season, and economic conditions. However, as described later in this paper, since the analysis is based on cross-sectional data for a specific region, the set of recipes is fixed throughout the analysis.

### 2.2. Data Used and Methods for Measuring Each Type of Complementarity

This paper measures the scales of the three complementarity concepts explained in the previous subsection for each combination of two foods among the 95 food items included in the FFQ (or 82 items in the alternative measurement method described later), based on their respective data. The three complementarity scales are calculated for each of the $\left(\begin{array}{c} 95\\ 2 \end{array}\right)=$ 4465 combinations of foods (or $\left(\begin{array}{c} 82\\ 2 \end{array}\right)=$ 3321 in the alternative measurement method).

In addition, when measuring the three types of complementarity scales for each combination of two foods, if there is variability in the parameters of the measurement method (such as the complementarity in habits with conditional variables X in Equation (5)), the measurements are conducted based on the following four sub-populations: (1) females aged 50 and above, (2) females aged 20 to 49, (3) males aged 50 and above, and (4) males aged 20 to 49. The data and measurement methods used for each of the three complementarity concepts are sequentially explained below.

#### 2.2.1. Measuring Nutrient Complementarity

As mentioned above, the nutrient complementarity of the generic combination of two foods $\left(i,\;j\right)$ is defined on the normalized nutrient content vectors $\left(z_{i},\;z_{j}\right)=S^{-1}\left(x_{i},\;x_{j}\right)$, where the matrix S is the sample standard deviation matrix of the nutrient content vectors. The data for the nutrient content vector $x_{i}$ are derived from the Standard Tables of Food Composition in Japan [9]. For those unfamiliar with the standard deviation matrix, the calculation example is provided in Appendix A.

In this paper, to measure the nutrient content vectors per unit volume of food, we alternatively use the following two units: (a) the standard intake per serving according to the FFQ and (b) the equivalent of 100 JPY. The 100 JPY equivalent is calculated using the retail price data from Sapporo City in the Retail Price Survey (Trend Survey) [10]. The prices for each food item are average prices from 2014 to 2018, except for some exceptional food items. The reason this paper uses the two types of nutrient content per unit quantity mentioned above, instead of the commonly used nutrient content per unit energy intake found in diet evaluation literature, is to take into account dietary practices and economic conditions as decision-making factors in people’s behavioral choices. Criterion (a) represents the most straightforward and convenient approach, based on the portion size assumed in the FFQ as the standard intake per meal for the average Japanese individual. However, naturally, meals are not always prepared in such standard portion combinations. Depending on the variety of dishes, considerable variation is expected in the combinations of portion sizes for each set of foods. Criterion (b), on the other hand, is based on monetary cost, appropriately reflecting the effect where expensive ingredients are consumed in smaller quantities compared to inexpensive ones. Of course, recipes may not always take into account financial considerations. However, it is somewhat acceptable for analysts to anticipate a filtering effect within the communication space, whereby recipes that are extremely indifferent to ingredient costs may be naturally excluded. In method (a), the complementarity of nutrients is measured for each combination of two items among the 95 food items. However, in method (b), due to the availability of Retail Price Survey data, it is measured for each combination of two items among 82 out of the 95 food items. Table A1 shows the FFQ standard intake and the amount equivalent to 100 JPY for each food item analyzed. Refer to [11] for details on the Japanese version of the FFQ.

For the dimensions of the nutrient content vector $x_{i}$, we selected nutrients for each of the above-mentioned four sub-populations from the 26 items (excluding energy, carbohydrate, and sodium) published in the National Health and Nutrition Survey [12], which provide national average intake values and their standard deviations by gender and age group. Among these, only the nutrients for which the number of cohorts rejecting the null hypothesis “mean < Dietary Reference Intake (DRI) value” (the sign is reversed for saturated fatty acids) in surveys from 2012 to 2019 is zero were included in the assessment of nutrient complementarity. Table A2 shows the number of cohorts for which the null hypothesis was rejected in tests of the differences between the mean nutrient intake and the DRI values with the National Health and Nutrition Survey data.

#### 2.2.2. Measuring Complementarity in Habits

To estimate the complementarity in habits for each combination of foods, this paper utilizes observational study data. While the survey itself was conducted from the summer of 2019 to the winter of 2020, among the extensive range of survey items, this paper uses the FFQ data collected in the winter of 2020. The survey participants were males and females aged in their 20s to 70s living in five prefectures of Japan who were not suffering from severe illnesses. For the analysis in this paper, however, the sample with the largest size, from Ebetsu City in Hokkaido, is used. A detailed explanation of the study protocol has been previously reported [13].

This study was approved by the Ethics Committee of the Hokkaido Information University (approval date: 22 April 2019; approval number: 2019-04), and written consent was obtained from the participants. The research was conducted in accordance with the Helsinki Declaration.

The complementarity in habits is calculated for each of the aforementioned four sub-populations. The complementarity $p^{habit}$ in Equation (5) is estimated using the same method and data as in our previous study [8]. In the previous study, the sample was divided into two groups based on gender for estimation; however, in this paper, the sample is divided into four groups based on gender and age category (whether they are aged 50 or older, or under 50) for estimation. The sample size and mean age ± standard deviation for each of the four sub-samples used in the analysis are as follows: (1) females aged 50 and above: N = 324, age = 58.3 ± 6.2; (2) females aged 20 to 49: N = 266, age = 40.2 ± 7.4; (3) males aged 50 and above: N = 128, age = 61.8 ± 7.0; and (4) males aged 20 to 49: N = 82, age = 40.6 ± 7.1.

#### 2.2.3. Measuring Complementarity in Recipes

We estimated the complementarity in recipes using the Cookpad dataset provided by Cookpad Inc. via the IDR Dataset Service of National Institute of Informatics [14]. The Cookpad is one of the most well-known recipe search sites in Japan. We calculated the frequency of ingredient combinations using the data of ingredient lists for each recipe included in the dataset.

The Cookpad database used in this paper includes approximately 1.7 million recipes submitted between around the year 2000 and around 2014. Since the recipe data are text data written freely by the submitters, we cleaned the ingredient name text data before using them for analysis. The details of the data wrangling for the analysis of this paper using Cookpad data are described in Appendix A.

### 2.3. Analysis

We will examine whether there is room for improvement in diet using recipes for four sub-populations of healthy individuals in Ebetsu City, Hokkaido, divided by gender and age group. Here, “room for improvement in diet using recipes” can be formulated using three concepts of complementarity related to food combinations. That is, for a person, the room for improvement in diet using recipes is a set of food combinations that (i) have high complementarity of nutrients, (ii) have low habit complementarity, and (iii) have high recipe complementarity. These are combinations of foods that, despite efficiently supplying nutrients to the person, are not actually consumed together, and it is expected that by utilizing recipes, these food combinations can be actively incorporated into the person’s diet.

Here, for the purpose of analysis, we define two types of sets of food combinations based on the concepts of complementarity. First, for a person, a set of combinations of foods with (i) high complementarity of nutrients and (ii) low habit complementarity represents, so to speak, “adjacent unexplored land” or “marginal virgin land” in terms of improving their diet. We denote this set of τ×100% margin by $M\left(\tau \right)$:(7)$$M\left(\tau \right)\equiv \left\{\left\{i,\;j\right\}|\begin{array}{ccc} \begin{array}{c} \theta \left(z_{i},\;z_{j}\right)\geq \theta_{1-\tau }\\ \eta \left(z_{i},\;z_{j}\right)\geq \eta_{1-\tau }\\ \varphi \left(z_{i},\;z_{j}\right)\geq \varphi_{1-\tau } \end{array} & \& & p^{habit}\left(i,\;j\right)\leq \pi_{\tau }^{habit} \end{array}\right\}\;=\left\{\left\{i,\;j\right\}|\begin{array}{ccc} I\left(z_{i},\;z_{j}\right)\geq 1-\tau & \& & p^{habit}\left(i,\;j\right)\leq \pi_{\tau }^{habit} \end{array}\right\}\;\left(0\leq \tau \leq 1\right),$$
where $\theta_{1-\tau }$, $\eta_{1-\tau }$, and $\varphi_{1-\tau }$ are the $\left(1-\tau \right)\times 100$th percentiles of their respective distributions, and $\pi_{\tau }^{habit}$ is the τ×100th percentiles of the distribution of $p^{habit}$. In Equation (7), the parameter τ corresponds to the “level of challenge in frontier development”, with lower values indicating a higher degree of “level of challenge”, i.e., higher complementarity of nutrients and lower habit complementarity.

Second, the set of the top τ×100% combinations of food items that are frequently used together in the same recipe is represented by $R\left(\tau \right)$ as follows:(8)$$R\left(\tau \right)\equiv \left\{\left\{i,\;j\right\}|p^{recipe}\left(i,\;j\right)\geq \pi_{1-\tau }^{recipe}\right\}\;\;\;\;\left(0\leq \tau \leq 1\right),$$
where $\pi_{1-\tau }^{recipe}$ is the $\left(1-\tau \right)\times 100$th percentiles of the distribution of $p^{recipe}$.

The main focus of this paper’s analysis is the intersection of set $M\left(\tau \right)$ and set $R\left(\tau \right)$, denoted as $\pi_{1-\tau }^{recipe}$. To express it metaphorically, the improvement of food is like the exploration of a frontier, where set $M\left(\tau \right)$ represents adjacent uncharted territory or marginal virgin land, and set $R\left(\tau \right)$ represents accessibility, such as the ease of laying roads.

### 2.4. Computer Processing

In the analysis presented in this paper, data processing was conducted using Python (version 3.11.5).

## 3. Results

As mentioned earlier, in this paper’s analysis, we calculated the complementarities in three aspects: complementarities of nutrients, habit complementarities, and recipe complementarities, for four groups: (1) females aged 50 and above, (2) females aged 20 to 49, (3) males aged 50 and above, and (4) males aged 20 to 49. These calculations were performed for (a) 4465 food combinations based on the FFQ standard intake criteria and (b) 3321 food combinations based on the 100 JPY equivalent criteria. We obtained a total of eight patterns of analysis results (4 groups × 2 criteria). In the main text, unless otherwise specified, we focus on the results of the two criteria, (a) and (b), for the group of females aged 50 and above, while the results for the remaining groups are presented in the Appendix A.

## 3.1. Measurement Results of Nutrient Complementarity

Figure 1 shows the results of calculating the complementarity of nutrients for each combination of foods, based on the nutrient content per standard amount of the FFQ and per 100 JPY equivalent, with respect to nutrients that are often insufficient in females aged 50 and above in Japan. In the figure, the color scale represents the magnitude of composite measure $I\left(z_{i},\;z_{j}\right)$ in Equation (4), with values increasing from blue to red. The results for the other groups (females aged 20 to 49, males aged 50 and above, males aged 20 to 49) are shown in Appendix A. Since the list of nutrients that tend to be deficient for each group (Table A2) does not dramatically differ across the sub-populations, the scatter plots showing the distribution of nutrient complementarity do not appear to reveal substantial differences between the groups.

In Figure 1, one point that should be noted is the systematic relationship between the quantitative effect η and the symmetry concerning quantity φ. The greater the absolute values of the standardized nutrient vectors of both combined foods, the greater the quantitative effect η; however, in this case, the symmetry of the two foods also inevitably increases. Therefore, the scatter plot on the φ-η plane will show a protruding shape extending upward to the right.

Table 1 shows the top five and bottom five food combinations for the composite measure I based on complementarity shown in panel (a) of Figure 1. The top combinations have values that exceed the 90th percentile for θ, η, and φ. In the top combination of pacific saury and mackerel and eel, among the six analyzed nutrients (retinol, vitamin D, vitamin B6, K, Ca, Mg, and Zn), vitamin B6 is abundant in pacific saury and mackerel, but scarce in eel, while retinol is abundant in eel, but scarce in pacific saury and mackerel. This results in a high value of the narrow-sense complementarity θ. Additionally, because the nutrient content per standard amount of FFQ is relatively high in both foods, the quantitative effect η and the symmetry regarding quantity φ also exhibit high values. The bottom combinations often have high values for φ, but small values for θ or η, resulting in a low I. An exceptional case in the bottom five is the combination of liver and pickled plum, which has high values for θ and η, but a low value for φ (symmetry), resulting in a low I.

## 3.2. Potential for Effortless Dietary Improvements Using Recipes

In the following analysis, outliers in the distribution of complementarity in habits were observed in all four gender- and age-based groups, and a clear cluster structure was observed in the joint distribution with nutrient complementarity and recipe complementarity (Appendix A). Therefore, we will focus our analysis on the main cluster that includes approximately 90% of the samples, both (a) based on the FFQ standard intake criteria and (b) based on the 100 JPY equivalent criteria.

Figure 2 shows the relationship between the number of elements in set $M\left(\tau \right)$ in Equation (7) and set $M\left(\tau \right)\cap R\left(\tau \right)$ in Equations (7) and (8), and the parameter τ based on the configuration of females aged 50 and above (see Appendix A for the other groups). The horizontal axis represents τ, and the vertical axis represents the number of food combinations. The blue line indicates the number of elements in set $M\left(\tau \right)$, and the red line represents the number of elements in set $M\left(\tau \right)\cap R\left(\tau \right)$.

In the explanation of Figure 2, we would like to use the metaphor of “cultivating uncharted land”, which was also employed in the explanation of Equations (7) and (8). The blue lines in Figure 2 represent the breadth of the “food improvement possibility frontier” according to the value of parameter τ. The lower the value of parameter τ, the greater the level of challenge of frontier development, in other words, higher complementarity of nutrients and lower habit complementarity. The farthest point on the frontier is at τ = 0.13 for both (a) FFQ standard amount and (b) 100 JPY equivalence. Additionally, the red line in Figure 2 represents the range of the food improvement possibility frontier that can be expected to be developed through recipes. In other words, it is the set of food combinations with high complementarity of nutrients, low habit complementarity, and high recipe complementarity. The farthest point that can be developed through recipes is at τ = 0.24 for both (a) FFQ standard amount and (b) 100 JPY equivalence.

Based on the results of Figure 2, we examined the food improvement potential frontier $M\left(\tau \right)$ and the recipe improvement potential area $M\left(\tau \right)\cap R\left(\tau \right)$ with *τ* = 0.3. Figure 3 shows the distribution of set $M\left(\tau \right)$ and set $M\left(\tau \right)\cap R\left(\tau \right)$ in the space of the three complementarity measures for food combinations. Each point represents a combination of two foods, with blue markers representing set $M\left(\tau \right)$ and red markers representing set $M\left(\tau \right)\cap R\left(\tau \right)$.

For all four groups classified by gender and age, the elements of the set *M*∩*R* are shown in Table 2. For female aged 50 and over, (a) according to the FFQ standard intake criteria, milk and pasta, and salmon and trout and pasta were listed, and (b) according to the 100 JPY equivalent amount standard, milk and pasta, salmon and trout and pasta, horse mackerel and sardine and carrots, and onions and taro were listed. Incorporating recipes using these food combinations into dietary habits is an easily implementable dietary improvement for this sub-population.

In Table 2, the results for (a) criteria for females under 50 and (b) criteria for men over 50 are not displayed, as both are empty sets. Thus, depending on the target group and the value of parameter τ, the set $M\left(\tau \right)\cap R\left(\tau \right)$ may become an empty set. This implies that there is no longer any room for dietary improvement using recipes for these groups.

## 4. Discussion

In the context of promoting healthy eating, the importance of considering dietary patterns as combinations of foods consumed simultaneously has been pointed out [15]. This paper focuses on combinations of two foods consumed simultaneously as the basic units of dietary patterns. By evaluating the complementarity (i) of nutrients, (ii) in habits, and (iii) in recipes for each combination of two foods, this paper presents a framework for discussing the potential for improving diet based on current prevailing dietary habits.

There are studies that consider dietary patterns as combinations of arbitrary numbers of foods (e.g., [15,16]). The analysis in this paper limits the number of foods in combinations to two due to computational constraints. However, conceptually, the analytical framework of this paper can be extended to combinations of three or more foods. Compared to previous studies, including those using linear programming methods for improving diets (e.g., [17,18,19,20,21,22,23,24,25,26,27,28,29,30]), the unique aspects of this paper are primarily its explicit consideration of individuals’ ingrained habits as constraints in dietary improvement and its discussion on the potential use of recipes as a tool. Although linear programming is rigorous, yet clear, easy to understand, and widely applicable, it is not without its weaknesses when applied to the problem of dietary improvement. In the analysis of dietary improvement using linear programming, deviations from observed dietary patterns are minimized under the constraints of nutritional standards and feasibility criteria. Here, the minimization of deviations and the constraints of feasibility criteria reflect consideration for established dietary habits, while the constraints of nutritional standards reflect consideration for health. However, in the minimization of deviations, for example, a 1% change in vegetable intake and a 1% change in meat intake are evaluated with the same weight, but whether this is appropriate as an evaluation of the burden of dietary improvement is not self-evident. Furthermore, the results obtained from linear programming often include corner-point solutions, which may pose significant challenges in the practical implementation of dietary habits. In contrast, the method proposed in this paper is expected to mitigate excessive practical burdens through the high complementarity in recipes.

Additionally, the formulation and measurement methods for complementarity concepts in this paper are also distinctive. In a word, the analytical framework of this paper states that if there are combinations of foods with (i) high complementarity of nutrients and (ii) low habit complementarity, but with (iii) sufficiently high recipe complementarity, then improving diet through recipes using these food combinations can achieve both efficient nutrient intake and a reduction in psychological burden on individuals during dietary improvement. As mentioned in the introduction, it is assumed that individual established habits are malleable. The reformation of habits can be more psychologically challenging or easier for individuals, and this challenge can be alleviated by some tools. In this paper, the tool in question is the use of recipes. The dietary improvement advice provided by the analytical framework of this paper takes the following form: “If a group with certain attributes (in this paper, specific region, gender, and age group) intends to make further improvements to their current eating habits, it is advisable to consciously incorporate recipes that use combinations of Food 1 and Food 2, or Food 3 and Food 4 as ingredients into their daily life.” The analytical framework is also expected to be utilized in the digitalization of dietary management, which has seen remarkable advancements in recent years [31,32]. In this study, the analysis on dietary habits relies on observational research with a sample size of approximately 800, whereas relatively large-scale data are utilized for recipes. The calculation of complementarity in habits is limited to specific regions, with conditions constrained solely by gender and age group. If, in the future, big data can also be applied to dietary habits, broader conditioning could enable the analytical methods of this study to yield more generalized implications for dietary improvement.

Of course, the simultaneous consumption of two foods does not only occur through recipes. Two foods can be consumed simultaneously as ingredients in a single recipe, but also as ingredients in different recipes. In this regard, it is important to note that dietary improvement using recipes as tools represents only a part of the potential for improvement.

Among the food combinations in recipes suggested as potentially useful for dietary improvement in the analysis results (Table 2), there are occasionally combinations of foods that might seem somewhat unexpected. For example, the combination of milk and pasta might not align with the typical image of Japanese cuisine. However, dietary diversification has already advanced significantly in Japan. Moreover, the majority of the recipes included in our dataset were proposed within Japan.

This paper examines the potential for dietary improvement by gender and age group using FFQ data from males and females without significant health issues in a specific region of Japan. Given that the deficiencies in nutrients and acquired habits differ by gender and age group, this analysis takes those variations into account. As such, it can be seen as a pioneering attempt at personalized nutritional guidance that addresses individual challenges and living environments.

The limitations of this paper are as follows. First, as a means of approaching a healthy dietary pattern, the combination intake of foods highlighted in this paper is evaluated based on the complementarity of nutrients, which is distinct from what might be called “nutritional complementarity.” Specifically, the combination intake of foods is not only about one food compensating for the nutrients lacking in another, but also about the fact that the nutrients in one food and different nutrients in another food can have a synergistic effect on health [33]. The method in this paper excludes consideration of this synergistic effect.

Second, the sub-population groups for dietary improvement were defined by gender and age class, and the potential for dietary improvement within each group was discussed. However, the sample data used are skewed toward a relatively healthy population. As shown in Appendix A, the sample generally meets dietary intake standards. Therefore, this analysis defines the set of nutrients that are often deficient for each gender and age group by utilizing published data from the National Health and Nutrition Survey. If the data used to estimate habits complementarity were also used to measure complementarity of nutrients, a trivial result would be obtained, showing no room for dietary improvement using recipes.

Third, due to data constraints, traditional Japanese staple foods such as rice, miso soup, and alcohol were not included in the analysis. Carbohydrates were excluded in the measurement of complementarity of nutrients because rice, a major carbohydrate source, was not included in the analysis. Sodium was also excluded for the same reason (i.e., miso soup excluded).

Fourth, in measuring complementarity of nutrients, two alternative standards were used for the unit quantity of food to define the nutrient content of each food: (a) FFQ standard intake and (b) the equivalent of 100 JPY. However, other candidates for unit quantities are conceivable, and there is no single correct answer. In particular, the intake per meal varies among individuals. Therefore, there are still issues to be considered for the measurement of complementarity of nutrients depending on the purpose and conditions of the analysis.

Fifth, dietary habits are influenced by various factors. However, in this paper, we estimated complementarity in dietary habits by conditioning only on gender and age. It is true that our observational data include demographic information such as educational background, occupation, and household members. Nevertheless, in practical terms, increasing the number of conditioning variables reduces the sample size. For this reason, in this paper, we used only gender and age group as the minimal information necessary to identify nutrients that are likely to be insufficient within specific sub-populations.

Finally, this analysis utilizes FFQ data; however, as it is self-reported and relies on memory, it has weaknesses such as recall bias, response bias, social desirability bias, and misclassification ([34] and its references). This paper estimates complementarity in habits using FFQ data, following the method described in [8]. The results of simulations assessing the impact of biases related to FFQ data on these estimates are presented in Appendix A. According to the simulation results, if the probability of response bias occurring is sufficiently low (10% or less), or if there is no correlation in the occurrence of response biases for the two food items, the bias in estimating the habit complementarity parameter will be moderate.

## 5. Conclusions

Improving diet in a healthy and practically feasible way requires considering the balance between the complementarity of nutrients and the complementarity in prevailing habits. Simultaneously consuming multiple foods with different functions is required for the complementarity of nutrient content, while it is constrained by individual acquired habits. This is where the analytical framework of this paper suggests there is room for the use of recipes. One example derived from the analysis results demonstrates the practical implications of the analytical method presented in this paper: If women aged 50 and above living in this region intend to make further improvements to their current dietary habits, it is recommended to consciously incorporate recipes that use combinations such as milk and pasta, salmon and pasta, or horse mackerel and sardines with carrots as ingredients.

## Figures and Tables

**Figure 1 nutrients-17-01614-f001:**
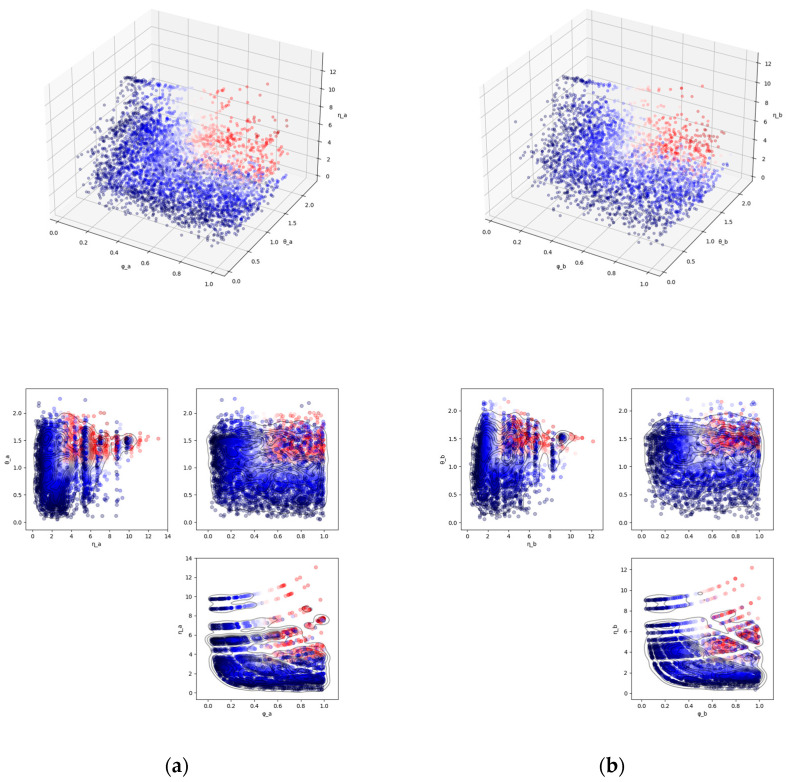
The distribution of nutrient complementarity composed of three elements $\left(\theta ,\;\eta ,\;\varphi \right)$. This is based on the calculation of nutrients that are often insufficient in females aged 50 and above in Japan (see Table A2). Each point corresponds to a combination of two foods, with the color scale representing the magnitude of the composite measure I in Equation (4). (**a**) Standard intake per serving according to the FFQ; (**b**) 100 JPY equivalent.

**Figure 2 nutrients-17-01614-f002:**
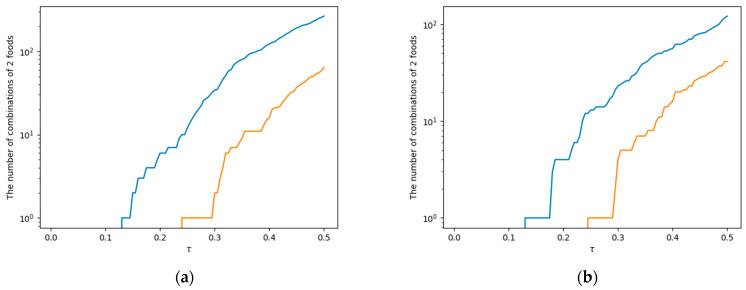
The dependence of the size of $M\left(\tau \right)$ (blue line) and the size of $M\left(\tau \right)\cap R\left(\tau \right)$ (red line) on the parameter τ (horizontal axis) for the group of females aged 50 and above. (**a**) Standard intake per serving according to the FFQ; (**b**) 100 JPY equivalent.

**Figure 3 nutrients-17-01614-f003:**
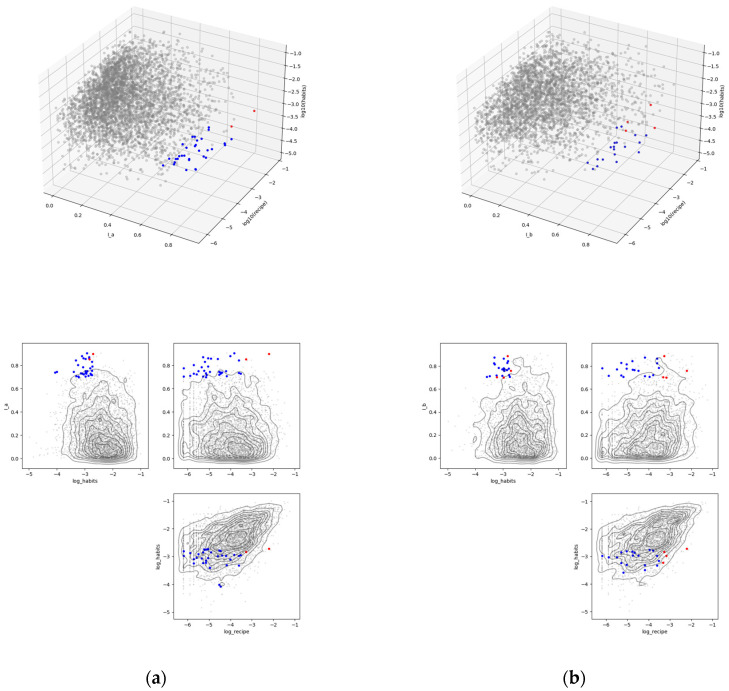
The sets $M\left(\tau \right)$ and $M\left(\tau \right)\cap R\left(\tau \right)$ in $\left(I,\;\log_{10}p^{habit},\;\log_{10}p^{recipe}\right)$ space for τ=0.3 for the group of females aged 50 and above. Blue markers are set $M\left(\tau \right)$, and red markers are set $M\left(\tau \right)\cap R\left(\tau \right)$. (**a**) Standard intake per serving according to the FFQ; (**b**) 100 JPY equivalent.

**Table 1 nutrients-17-01614-t001:** The top five and bottom five composite measure *I*.

Food *i*	Food *j*	$\left|z_{i}\right|$	$\left|z_{j}\right|$	θ	η	φ	$F_{\theta }$	$F_{\eta }$	$F_{\varphi }$	I
Pacific saury and mackerel	Eel	5.48	5.45	1.81	7.72	1.00	0.9772	0.9507	0.9971	0.9507
Tunas and bonito	Eel	5.05	5.45	2.00	7.42	0.95	0.9978	0.9409	0.9489	0.9409
Eel	Japanese noodles (soba)	5.45	5.26	1.65	7.57	0.98	0.9310	0.9458	0.9756	0.9310
Milk	Japanese noodles (soba)	5.55	5.26	1.60	7.64	0.97	0.9068	0.9485	0.9622	0.9068
Pacific saury and mackerel	Spinach	5.48	5.34	1.60	7.65	0.98	0.9057	0.9487	0.9837	0.9057
⋮										⋮
Strawberry	Eggplant	1.78	1.88	0.06	2.59	0.97	0.0004	0.3805	0.9617	0.0004
Pickled plum	Sesame	0.15	0.22	1.36	0.27	0.77	0.7236	0.0004	0.7774	0.0004
Peach	Pears	1.86	1.76	0.05	2.57	0.96	0.0002	0.3711	0.9608	0.0002
Liver	Pickled plum	9.71	0.15	1.48	9.72	0.02	0.8119	0.9765	0.0002	0.0002
Pickled plum	Scallion	0.15	0.22	0.73	0.27	0.78	0.2432	0.0002	0.7948	0.0002

Note: List of the top five and bottom five composite measure *I* of nutrient complementarity based on nutrients that are often deficient in females aged 50 and above (see Table A2). The food unit criterion is (a) the FFQ standard amount. $F_{\theta }$, $F_{\eta }$, and $F_{\varphi }$ are the empirical cumulative distribution function of θ, η, and φ, respectively. The ellipsis in the table indicates that records in the middle rankings, between the top five and bottom five, have been omitted.

**Table 2 nutrients-17-01614-t002:** The elements of $M\left(\tau \right)\cap R\left(\tau \right)$ for τ=0.3.

(1) Females Aged 50 and Above													
	food i	food j	θ	η	φ	$F_{\theta }$	$F_{\eta }$	$F_{\varphi }$	I	$\log_{10}p^{habit}$	${\bar{F}}_{h}$	$\log_{10}p^{recipe}$	$F_{recipe}$
(a)	Milk	Pasta	1.59	7.77	0.99	0.90	0.95	0.99	0.90	−2.71	0.70	−2.21	0.98
	Salmon and trout	Pasta	1.57	8.86	0.84	0.89	0.97	0.85	0.85	−2.84	0.76	−3.27	0.77
(b)	Milk	Pasta	1.63	7.91	0.77	0.89	0.95	0.76	0.76	−2.71	0.70	−2.21	0.97
	Salmon and trout	Pasta	1.84	6.27	0.98	0.98	0.89	0.98	0.89	−2.84	0.76	−3.27	0.76
	Horse mackerel and sardine	Carrot	1.44	4.74	0.78	0.71	0.77	0.77	0.71	−3.23	0.91	−3.33	0.74
	Onion	Taros	1.48	4.62	0.72	0.75	0.75	0.70	0.70	−2.98	0.82	−3.17	0.79
**(2) Females aged 20 to 49**													
	food i	food j	θ	η	φ	$F_{\theta }$	$F_{\eta }$	$F_{\varphi }$	I	$\log_{10}p^{habit}$	${\bar{F}}_{h}$	$\log_{10}p^{recipe}$	$F_{recipe}$
(b)	Beef	Enoki mushroom	1.72	6.98	0.97	0.94	0.80	0.96	0.80	−3.41	0.85	−3.35	0.71
	Clam and corb shell	Potatoes	1.56	10.84	0.91	0.81	1.00	0.89	0.81	−3.58	0.90	−3.37	0.70
**(3) Males aged 50 and above**													
	food i	food j	θ	η	φ	$F_{\theta }$	$F_{\eta }$	$F_{\varphi }$	I	$\log_{10}p^{habit}$	${\bar{F}}_{h}$	$\log_{10}p^{recipe}$	$F_{recipe}$
(a)	Salmon and trout	Pasta	1.58	8.85	0.88	0.87	0.95	0.87	0.87	−3.01	0.81	−3.27	0.75
**(4) Males aged 20 to 49**													
	food i	food j	θ	η	φ	$F_{\theta }$	$F_{\eta }$	$F_{\varphi }$	I	$\log_{10}p^{habit}$	${\bar{F}}_{h}$	$\log_{10}p^{recipe}$	$F_{recipe}$
(a)	Cod roe and salmon roe	Potatoes	1.79	7.40	0.97	0.98	0.78	0.97	0.78	−3.12	0.73	−2.95	0.82
	Clam and corb shell	Pasta	1.49	11.20	0.84	0.76	0.99	0.83	0.76	−3.26	0.80	−3.05	0.79
(b)	Milk	Salmon and trout	1.51	8.87	0.85	0.70	0.92	0.81	0.70	−3.94	0.97	−3.03	0.79

Note: The leftmost column shows two criteria for measuring the amount of contained nutrients: (a) FFQ standard intake and (b) 100 JPY equivalent. $F_{\theta }$, $F_{\eta }$, $F_{\varphi }$, and $F_{recipe}$ are the empirical cumulative distribution function of θ, η, φ, and $\log_{10}p^{recipe}$, respectively. ${\bar{F}}_{h}$ is the empirical complementary cumulative distribution function of $\log_{10}p^{habit}$.

## Data Availability

The data obtained from the “*Sukoyaka* Health Survey” are available in a publicly accessible repository managed by the DNA Data Bank of Japan (DDBJ) Japanese Genotype Phenotype Archive at https://www.ddbj.nig.ac.jp/jga/index-e.html, accessed on 18 February 2025.

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
