# Peer review of "Easily Implementable Dietary Improvement Using Recipes: Analytical Method Applied to a Specific Region in Japan"

_nutrients, 2025, doi:10.3390/nu17101614_

Round 1

Reviewer 1 Report

Comments and Suggestions for Authors
  1. This study investigated three concepts of complementarity (nutrition, habits, and recipes) and their potential influence on dietary behaviors. While the manuscript provides an interesting exploration of these components, it would benefit from a clearer articulation of how these elements interact to shape the dietary practices. Moreover, the distinct and combined explanatory power of these concepts in elucidating the observed dietary patterns warrants further elaboration. Providing such clarification would enhance the theoretical contributions of this research and help establish its unique value within the field of dietary improvement.
  2. The literature review requires considerable expansion to situate the study within the broader context of research on dietary patterns, food combinations and nutritional interventions. Key areas that need to be addressed include the role of food combinations in enhancing nutrient bioavailability and dietary diversity, the behavioral and cultural determinants of dietary habits and how these interact with nutrition-based interventions, and previous studies on the measurement and impact of habitual dietary behaviors in different populations.
  3. The study employed the FFQ to measure dietary intake and utilized the 100-yen cost equivalent metric to assess nutritional complementarity. Although these methodological approaches are valid, a more detailed justification for their selection is necessary. The manuscript should explicitly discuss the rationale behind these choices and weigh their respective strengths and limitations. This will enhance the transparency and robustness of the methodological framework.
  4. The analysis would benefit from greater consideration of additional confounding variables that may significantly influence dietary patterns. Factors such as lifestyle behaviors (e.g., physical activity levels or time spent cooking), socioeconomic status (e.g., household income, food security), and educational background (e.g., nutritional literacy) should be statistically controlled for in the analysis or explicitly acknowledged as limitations of the study. Accounting for these variables would help to ensure the reliability and validity of the findings.
  5. As the data for this study were collected from a specific region in Japan, the findings may have limited generalizability to other cultural and demographic contexts. Including a critical discussion of how cultural, regional, and demographic variations might influence the applicability and relevance of the results would be beneficial. Such a discussion would enrich this paper and provide valuable insights for future research in other settings.
  6. The conclusion section could be strengthened by offering clearer and more actionable practical implications of the study’s findings. While the manuscript contributes to academic discourse, identifying specific recommendations for policymakers, practitioners, or researchers would enhance its impact and practical relevance.

Author Response

Comments 1: 
This study investigated three concepts of complementarity (nutrition, habits, and recipes) and their potential influence on dietary behaviors. While the manuscript provides an interesting exploration of these components, it would benefit from a clearer articulation of how these elements interact to shape the dietary practices. Moreover, the distinct and combined explanatory power of these concepts in elucidating the observed dietary patterns warrants further elaboration. Providing such clarification would enhance the theoretical contributions of this research and help establish its unique value within the field of dietary improvement.
Response 1: 
Thank you very much for your highly valuable suggestions. We have revised the Introduction, Discussion, and Conclusion sections.

Comments 2: 
The literature review requires considerable expansion to situate the study within the broader context of research on dietary patterns, food combinations and nutritional interventions. Key areas that need to be addressed include the role of food combinations in enhancing nutrient bioavailability and dietary diversity, the behavioral and cultural determinants of dietary habits and how these interact with nutrition-based interventions, and previous studies on the measurement and impact of habitual dietary behaviors in different populations.
Response 2: 
We have added several references to the manuscript. While there is an abundance of related literature, we chose to limit the citations to avoid blurring the focus of this paper. The most desired references for this paper were those related to the "practicality of food improvement measures at an individual level," but we were unable to find suitable ones.

Comments 3: 
The study employed the FFQ to measure dietary intake and utilized the 100-yen cost equivalent metric to assess nutritional complementarity. Although these methodological approaches are valid, a more detailed justification for their selection is necessary. The manuscript should explicitly discuss the rationale behind these choices and weigh their respective strengths and limitations. This will enhance the transparency and robustness of the methodological framework.
Response 3: 
We have expanded the explanation of the two criteria in 2.2.1. Measuring Nutrient Complementarity.

Comments 4: 
The analysis would benefit from greater consideration of additional confounding variables that may significantly influence dietary patterns. Factors such as lifestyle behaviors (e.g., physical activity levels or time spent cooking), socioeconomic status (e.g., household income, food security), and educational background (e.g., nutritional literacy) should be statistically controlled for in the analysis or explicitly acknowledged as limitations of the study. Accounting for these variables would help to ensure the reliability and validity of the findings.
Response 4: 
The broader the range of controllable confounding factors, the greater the generalizability of the analysis results. Therefore, we interpreted this comment, along with the following one, as addressing the so-called issue of missing variables. This point has been mentioned in the discussion section (... In this study, the analysis on dietary habits relies on observational research with a sample size of approximately 800, ...).

Comments 5: 
As the data for this study were collected from a specific region in Japan, the findings may have limited generalizability to other cultural and demographic contexts. Including a critical discussion of how cultural, regional, and demographic variations might influence the applicability and relevance of the results would be beneficial. Such a discussion would enrich this paper and provide valuable insights for future research in other settings.
Response 5: 
We understood this comment, along with the previous one, as relating to the issue of generalization. Please refer to our response to the previous comment.

Comments 6: 
The conclusion section could be strengthened by offering clearer and more actionable practical implications of the study’s findings. While the manuscript contributes to academic discourse, identifying specific recommendations for policymakers, practitioners, or researchers would enhance its impact and practical relevance.
Response 6: 
We have added the practical implication of our framework to the conclusion section.

Reviewer 2 Report

Comments and Suggestions for Authors

Dear Authors,

The manuscript sent for review seems interesting.

The concepts of complementarity based on nutrients, eating habits, and recipes are somewhat novel. However, the article as a whole is incomprehensible to me. I do not quite understand the purpose of this convoluted analysis.

Should we treat it as a methodological article? If so, what are this calculation method's recommendations, advantages, and disadvantages?

Suppose we are to treat it as information about the nutrition of the Japanese population. In that case, I miss the characteristics of the Japanese population taking into account these calculations and specifying that, for example, 20% of the studied population belongs to a group whose nutrition is based exclusively on eating habits and traditional recipes, and this is a particular age group, with a predominance of, for example, women, etc. This would give us an idea of ​​what can be extracted from such results.

Because of individualizing nutrition, I think 24-hour interviews are enough; we will know what they like, where they make mistakes, and what corrections of nutrition should be made.

So in this case, the description of the results is very enigmatic to me.

The authors describe the methodology in great detail, but can other scientists use it? In my opinion, unfortunately, not.

Looking at the references, the authors also do not know how to use the available nutritional data published by other authors. It seems impossible that there were no articles of this type, unless we consider it a methodologically innovative article.

Author Response

Reviewer 2
Comments and Suggestions for Authors
Dear Authors,
The manuscript sent for review seems interesting.

Comments 1: 
The concepts of complementarity based on nutrients, eating habits, and recipes are somewhat novel. However, the article as a whole is incomprehensible to me. I do not quite understand the purpose of this convoluted analysis.
Response 1: 
Thank you vary much for your comments. As pointed out in Comment 2, this paper aims to present an analytical framework. To make this point clear, We have revised the entire draft accordingly.

Comments 2: 
Should we treat it as a methodological article? If so, what are this calculation method's recommendations, advantages, and disadvantages?
Response 2: 
As you pointed out, this paper pertains to a new analytical method. In the discussion section, we have added a more detailed discussion comparing it to the analytical framework using linear programming, which is one of the established methods.

Comments 3: 
Suppose we are to treat it as information about the nutrition of the Japanese population. In that case, I miss the characteristics of the Japanese population taking into account these calculations and specifying that, for example, 20% of the studied population belongs to a group whose nutrition is based exclusively on eating habits and traditional recipes, and this is a particular age group, with a predominance of, for example, women, etc. This would give us an idea of what can be extracted from such results.
Response 3: 
As stated in the response to Comment 2, the analytical framework is the main focus of this paper. The characteristics of the data are mentioned to the extent that the focus does not become blurred.

Comments 4: 
Because of individualizing nutrition, I think 24-hour interviews are enough; we will know what they like, where they make mistakes, and what corrections of nutrition should be made.
Response 4: 
The advantages of a 24-hour dietary behavior record are as you pointed out. However, the main focus of this paper is to present a method for exploring the possibility of practical dietary improvements by focusing on three complementarities. As pointed out in Comment 2, we have revised this paper with an awareness that its focus is on conceptualization and the presentation of measurement methods.

Comments 5: 
So in this case, the description of the results is very enigmatic to me.
Response 5: 
M represents the set of food combinations that have high nutritional complementarity but are not currently practiced, while R represents the set of food combinations frequently used together as ingredients in recipes. The set M constitutes the frontier for dietary improvements in the target population, and the intersection of the sets M and R, denoted as M∩R, represents a frontier that can be explored through recipes. The visualization of this intersection, M∩R, is a key point in the analytical findings of this paper.

Comments 6: 
The authors describe the methodology in great detail, but can other scientists use it? In my opinion, unfortunately, not.
Response 6: 
To enhance the reproducibility of the analytical method presented in this paper and for the convenience of a wider range of readers, Supplementary S3 have been added detailing the calculation procedure using numerical examples.

Comments 7: 
Looking at the references, the authors also do not know how to use the available nutritional data published by other authors. It seems impossible that there were no articles of this type, unless we consider it a methodologically innovative article.
Response 7: 
As pointed out in Comment 2, this is a methodological paper. We have revised the manuscript with that aspect in mind.

Reviewer 3 Report

Comments and Suggestions for Authors

This study proposes a dietary improvement framework based on the concept of triple complementarity, exploring feasible pathways to achieve healthy eating through recipe analysis using dietary data from residents in Ebetsu City, Hokkaido, Japan. The research establishes quantitative models for nutritional complementarity, dietary habit complementarity, and recipe complementarity, combining FFQ dietary survey data with Cookpad recipe big data to analyze different gender and age groups. While demonstrating innovation, the study requires supplementation of methodological details and deeper discussion to enhance academic rigor and practical value. 

Introduction

-It is recommended to elaborate on the theoretical framework of habit plasticity.

Materials and Methods

-The computational details of the standard deviation matrix 𝑆 should be explicitly described.

-It is recommended to supplement the key processing steps of the recipe data cleaning process.

-It is recommended to supplement specific measures for the anonymization of participants' data.

Results

-The food combinations in Table 1 should adopt standardized English nomenclature.

-It is recommended to optimize the chart interpretation.

Discussion

-The geographical representativeness of the sample (limited to healthy populations in a single Hokkaido city) and its potential impact on generalizability warrant further discussion.

-It is recommended to quantitatively analyze the effects of FFQ data recall bias on habitual complementary measurements.

-It is recommended to supplement the migration limitations of this method in cross-cultural scenarios.

-A direct comparison with linear programming-based dietary optimization methods should be explicitly detailed.

-Recommended to supplement references to the frontiers of digitalization of nutrition.

Author Response

Reviewer 3
Comments and Suggestions for Authors
Comments 1: 
This study proposes a dietary improvement framework based on the concept of triple complementarity, exploring feasible pathways to achieve healthy eating through recipe analysis using dietary data from residents in Ebetsu City, Hokkaido, Japan. The research establishes quantitative models for nutritional complementarity, dietary habit complementarity, and recipe complementarity, combining FFQ dietary survey data with Cookpad recipe big data to analyze different gender and age groups. While demonstrating innovation, the study requires supplementation of methodological details and deeper discussion to enhance academic rigor and practical value. 
Response 1: 
Thank you very much for your highly valuable suggestions. We have added the practical implications of our framework to the discussion section. The limitation of the concept of nutrition complementarity has also been added.

Comments 2: 
Introduction
-It is recommended to elaborate on the theoretical framework of habit plasticity.
Response 2: 
We have added three references.

Comments 3: 
Materials and Methods
-The computational details of the standard deviation matrix ? should be explicitly described.
Response 3: 
In response to Comment 5 below, we have added the Supplementary S3, where the calculation method for the standard deviation matrix is explicitly stated.

Comments 4: 
-It is recommended to supplement the key processing steps of the recipe data cleaning process.
Response 4:
We have included the data cleaning procedure in Supplementary S3.

Comments 5: 
-It is recommended to supplement specific measures for the anonymization of participants' data.
Response 5: 
We have added the Supplementary S3, where the calculation procedure for nutrient complementarity is demonstrated using numerical examples. For the calculation of complementarity in preferences, please refer to the cited reference [8]. As for the calculation of complementarity in recipes, we believe there are no particularly difficult points even for the general reader.

Comments 6: 
Results
-The food combinations in Table 1 should adopt standardized English nomenclature.
Response 6: 
The terminology used in Tables 1, 2, and A1 follows the English notation from a previous study that validated the Japanese version of the FFQ. The reference to the study has been added.

Comments 7: 
-It is recommended to optimize the chart interpretation.
Response 7: 
In the explanation of Figure 2, we have stated that, unlike the other figures, it uses the metaphor employed in Equations 7 and 8.

Comments 8: 
Discussion
-The geographical representativeness of the sample (limited to healthy populations in a single Hokkaido city) and its potential impact on generalizability warrant further discussion.
Response 8: 
The broader the range of controllable confounding factors, the greater the generalizability of the analysis results. Therefore, we interpreted this comment, along with the Comments 10, as addressing the so-called issue of missing variables. This point has been mentioned in the discussion section (... In this study, the analysis on dietary habits relies on observational research with a sample size of approximately 800, ...).

Comments 9: 
-It is recommended to quantitatively analyze the effects of FFQ data recall bias on habitual complementary measurements.
Response 9:
We have included the simulation analysis in Supplementary S9.

Comments 10: 
-It is recommended to supplement the migration limitations of this method in cross-cultural scenarios.
Response 10: 
We understood this comment, along with the Comments 8, as relating to the issue of generalization. Please refer to our Response 8.

Comments 11: 
-A direct comparison with linear programming-based dietary optimization methods should be explicitly detailed.
Response 11: 
We have added a comparison between linear programming and the method proposed in this paper to the discussion section.

Comments 12: 
-Recommended to supplement references to the frontiers of digitalization of nutrition.
Response 12: 
We have noted that the concepts of complementarities introduced in this paper are also expected to be utilized in the digitalization of dietary management.

Reviewer 4 Report

Comments and Suggestions for Authors

Dear Authors,

Thank you for your manuscript. Please find my comments below.

The paper is well-written but is presented somewhat unusually, using manual algebra calculations instead of specific statistical software.

In the Introduction, please provide a more detailed description of dietary habits and clearly explain their importance for health outcomes. Additionally, the study's aim(s) need to be explicitly stated.

Overall, the manuscript addresses an important public health topic, exploring nutrient deficiencies across different gender and age groups. However, the approach is heavily mathematical and lacks practical explanations and implications for practical application. I recommend expanding the discussion section to clearly explain and interpret your findings concerning potential health benefits or risks associated with the identified food combinations and any resulting nutrient deficiencies. While the statistical procedures and data normalization methods are described in adequate detail, their complexity may present challenges to some readers.

Author Response

Comments 1: 
The paper is well-written but is presented somewhat unusually, using manual algebra calculations instead of specific statistical software.
Response 1: 
Thank you very much for your highly valuable suggestions. We added Section 2.4, Computer Processing, to explicitly state that the processing was conducted using Python. Additionally, in response to requests from other anonymous reviewers, we added Supplementary S3 to demonstrate the calculation procedure for nutrient complementarity using numerical examples.

Comments 2: 
In the Introduction, please provide a more detailed description of dietary habits and clearly explain their importance for health outcomes. Additionally, the study's aim(s) need to be explicitly stated.
Response 2: 
While the purpose of this paper included some deeply explored content, it lacked a clearly stated perspective for broader understanding. Therefore, we have added an explanation in the introduction section. (The purpose of this paper, in short, is to present a method for finding highly practical food improvement measures.)

Comments 3: 
Overall, the manuscript addresses an important public health topic, exploring nutrient deficiencies across different gender and age groups. However, the approach is heavily mathematical and lacks practical explanations and implications for practical application. I recommend expanding the discussion section to clearly explain and interpret your findings concerning potential health benefits or risks associated with the identified food combinations and any resulting nutrient deficiencies. While the statistical procedures and data normalization methods are described in adequate detail, their complexity may present challenges to some readers.
Response 3: 
We have added the practical implications of our framework to the discussion section. The limitation of the concept of nutrition complementarity has also been added.

Round 2

Reviewer 2 Report

Comments and Suggestions for Authors

Dear Authors,

The revised manuscript submitted for re-review has many improvements. The authors have added to the manuscript's purpose to present a method for improving food quality in a diet. It is an interesting method, but I do not know if it is too complicated to find a practical application. However, in science, not everything can consistently be implemented.

Many corrections made by the authors to the manuscript have improved it.
References are still not appropriately cited (at the end of the manuscript).

It seems that the manuscript could be published in the Nutrients Journal.

Reviewer

Reviewer 3 Report

Comments and Suggestions for Authors

The authors have responded carefully to my comments, and have revised the manuscript accordingly, now the article is ready to be accepted.

Reviewer 4 Report

Comments and Suggestions for Authors

Dear Authors, 

Thank you for the improved version of your paper. Good job!